

# Determinants of antiretroviral therapy coverage in Sub-Saharan Africa

Fumitaka Furuoka[1] and Mohammad Zahirul Hoque[2]

[1] Asia-Europe Institute, University of Malaya, Kuala Lumpur, Malaysia
[2] Faculty of Medicine and Health Sciences, Universiti Malaysia Sabah, Kota Kinabalu, Sabah, Malaysia

## ABSTRACT

Among 35 million people living with the human immunodeficiency virus (HIV) in 2013, only 37% had access to antiretroviral therapy (ART). Despite global concerted efforts to provide the universal access to the ART treatment, the ART coverage varies among countries and regions. At present, there is a lack of systematic empirical analyses on factors that determine the ART coverage. Therefore, the current study aimed to identify the determinants of the ART coverage in 41 countries in Sub-Saharan Africa. It employed statistical analyses for this purpose. Four elements, namely, the HIV prevalence, the level of national income, the level of medical expenditure and the number of nurses, were hypothesised to determine the ART coverage. The findings revealed that among the four proposed determinants only the HIV prevalence had a statistically significant impact on the ART coverage. In other words, the HIV prevalence was the sole determinant of the ART coverage in Sub-Saharan Africa.

## INTRODUCTION

Universal access to antiretroviral therapy (ART) treatment was identified by the United Nations in the year 2000 as one of the Millennium Development Goals (MDGs). Antiretroviral therapy (ART) drugs are used to treat HIV in order to increase life expectancy of the infected individuals (*Walker & Hirsch, 2013*; *Volberding & Deeks, 2010*; *Nakagawa et al., 2012*). Following the initiative of the United Nations (UN) and the World Health Organization (WHO), considerable efforts have been made all over the world to achieve this ambitious goal by 2015 (*United Nations, 2015*).

According to statistics compiled by the Joint UN Programme on HIV/AIDS (UNAIDS), there were around 35 million people living with human immunodeficiency virus (HIV) in 2013. Among them, only 12.9 million people had access to ART treatment in the same year (*Joint United Nations Programme on HIV/AIDS, 2014a*). In order to ensure that universal access to ART treatment can be achieved, in 2013, the WHO modified its originally proposed guidelines and recommended that the treatment should be initiated in all patients with CD4 cell count 500 cells/mm$^3$ or less (*World Health Organization, 2013*).

Corresponding author
Fumitaka Furuoka,
fumitakamy@gmail.com

As an outcome of the global concerted efforts to promote ART treatment combined with several important findings concerning enhancing its effectiveness to increase survival rate of the infected persons (*Braitstein et al., 2006*; *Ray et al., 2010*; *Volberding & Deeks, 2010*; *Cohen et al., 2011*; *Deeks, 2013*; *Tanser et al., 2013*; *Vandormael et al., 2014*) the percentage of people living with HIV who were receiving ART treatment had drastically increased from 10% in 2006 to 37% in 2013. Overall, ART treatment has successfully reduced AID-related deaths by 7.6 million in the world, including 4.8 million in Sub-Saharan Africa since 1995 (*Joint United Nations Programme on HIV/AIDS, 2014a*). However, a major problem is that the ART coverage varies across countries and regions of the world. For example, in South Africa, the percentage of people living with HIV who received ART treatment was 42% in 2013. In Nigeria, only 20% of people living with HIV had access to the treatment (*Joint United Nations Programme on HIV/AIDS, 2014a*).

Despite the importance of this issue and the wide coverage it received in research literature, there is a lack of systematic empirical analyses of elements influencing ART coverage. Among several notable exceptions (*Lieberman, 2007*; *Nattrass, 2008*; *Schwardmann, 2008*; *Peiffer & Boussalis, 2010*; *Man et al., 2012*), a study by *Lieberman (2007)* examined socio-political determinants of ART coverage in 151 developing countries. The determinants included cultural fractionalisation, HIV prevalence, income level, government effectiveness and the level of democracy. It was found that three of these elements, namely, cultural fractionalisation, income level and government effectiveness had a statistically significant impact on ART coverage. Nattrass (*Lieberman, 2007*) analysed determinants of ART coverage in 82 developing countries. The study focused on such socio-economic and political variables as income level, foreign aid, political situation, HIV prevalence, the number of HIV patients in urban areas and the number of medical professionals. The researcher identified four elements, namely, foreign aid, HIV prevalence, the number of HIV patients in urban areas and the level of democracy among statistically significant determinants of ART coverage. *Schwardmann (2008)* estimated a comprehensive ART coverage model that included HIV prevalence, HIV patients in urban areas, income level, public health expenditure, the number of medical professionals and the amounts of foreign aid. Among these variables, HIV patients in urban areas and public health expenditure were found to be important determinants to affect ART coverage.

More recently, *Peiffer & Boussalis (2010)* examined determinants of ART coverage in 116 developing countries. Their estimation model included socio-political and economic variables, such as amounts of foreign aid, cultural fractionalisation, the level of urbanization, societal traditionalism and the level of democracy. Only two of the determinants proposed by the authors, namely, the level of international financing for HIV/AIDS programs and the income level, were found to have a significant impact on ART coverage. A study by *Man et al. (2012)* estimated a comprehensive ART model with seven hypothesized determinants, which comprised political governance, gender inequality, skilled birth attendance rate and disability-adjusted life years (DALY). The researchers applied this estimation model to all developing countries. They found that among the

proposed socio-political and economic determinants only political governance had a statistically significant impact on ART coverage.

Against such a backdrop, this study aims to identify determinants of ART coverage in 41 countries in Sub-Saharan Africa. In comparison with the previous investigations, the current study has two distinctive methodological features: firstly, the geographic scope and, secondly, the choice of variables. Regarding the geographical scope, there are vast discrepancies in socio-economic and public health conditions among developing countries. An empirical study with a very wide scope (e.g., one that includes all developing countries) would suffer from a problem of heterogeneity. In order to control heterogeneity of a sample, the current study focused on Sub-Saharan African countries only. This geographical scope is well justified because the most severe impact of HIV/AIDS epidemics has been on these countries. As UNAIDS statistics show, there is approximately 25 million people living with HIV in Sub-Saharan Africa and the area's share of global population living with HIV is 70% (*Joint United Nations Programme on HIV/AIDS, 2014b*).

Concerning the choice of variables, the current study assumes that only four elements had a decisive impact on ART coverage in Sub-Saharan Africa. These comprise HIV prevalence, level of national income, level of medical expenditure and number of nurses. Previous studies incorporated more than five elements in their estimation models including some rather abstract variables, such as cultural fractionalisation, societal traditionalism, level of democracy and political governance. Undeniably, such variables can have an impact on ART coverage in Sub-Saharan Africa. For example, a degree of societal traditionalism or the differences in political leadership style could affect levels of ART coverage. The problem is that the exact impact of such variables is not easy to assess. In other words, there are considerable methodological challenges in identifying and then unambiguously operationalizing variables concerning complex and often endogenous cultural concepts.

Another problem is the availability of data. For example, foreign aid can be one of important determinants of ART coverage. However, reliable data on foreign aid spent for ART coverage or for treatment of HIV patients in urban areas are lacking. To be more specific, information on the aggregate amount of foreign aid given to a certain recipient country is readily available. However, it is difficult to obtain reliable data on the amount of foreign aid spent on the promotion of ART treatment. Similarly, information on the number of HIV patients in the Sub-Saharan region is available. However, the exact number of HIV patients in rural areas is not known. For these reasons, the current study incorporates only the determinants that can be clearly and unambiguously defined and are supported by reliable data.

The data were obtained from the World Development Indicators database. This study employed both descriptive and inferential statistics. The descriptive statistics focused on general characteristics of ART coverage in Sub-Saharan African countries while the inferential statistics helped to identify determinants of ART coverage in the region. The data are at a national level and therefore they do not reflect heterogeneity relating

to ART coverage at a province or district level within countries. Despite this potential shortcoming, issues addressed in this study have important policy implications.

## DATA AND METHODS

This study assumes that four elements, namely, HIV prevalence, level of national income, level of medical expenditure and number of nurses, determine ART coverage in Sub-Saharan Africa. The relationship between ART coverage and the four hypothesized determinants can be expressed as:

$$ART = f(HIV^+, GDP^+, HED^+, NUR^+), \tag{1}$$

where *ART* is antiretroviral therapy coverage or the percentage of people living with HIV who have access to antiretroviral therapy, *HIV* is HIV prevalence or the percentage of people living with HIV in total population between the ages 15 and 49, *GDP* is income level or the real Gross Domestic Product (GDP) per person based on purchasing power parity (PPP) calculation, *HED* is the level of health expenditure or the real health expenditure per person based on PPP calculation, *NUR* is the number of nurses and midwives per one thousand people.

It should be noted that there are vast discrepancies in the level of inflation and purchasing power of local currencies among the 41 Sub-Saharan African countries in this study. Therefore, the *GDP* and *HED* variables represent 'real' values rather than 'nominal' values. 'Real' values are calculated based on 'nominal' values and they take account of changes in the inflation rate. These real values were measured in this study in terms of 2011 price levels in relevant countries. Furthermore, the *GDP* and *HED* were denominated in the International dollar rather than local currencies. International dollar denominated values were derived from a local currency denominated value so that differences in the purchasing power of local currencies were taken into account. The International dollar (I$) has the same purchasing power in a relevant country as the US dollar (US$) in the United States.

The World Development Indicators database, from which the data were obtained, categorises 48 countries in Africa as Sub-Saharan countries. The present study excluded seven countries, namely, Comoros, Equatorial Guinea, Mauritania, Seychelles, Somalia, South Sudan and Zimbabwe, from the analysis due to a lack of data. The World Development Indicators database codifies *ART* as SH.HIV.ARTC.ZS while *HIV* is denoted as SH.DYN.AIDS.ZS. These HIV related data were estimated by the UNAIDS. Next, *HED* is codified in the database as SH.XPD.PCAP.PP.KD and *NUR* is codified as SH.MED.NUMW.P3. These public health related data were estimated by the WHO. Finally, the database codifies *GDP* as NY.GDP.PCAP.PP.KD. The socio-economic data were estimated by the *World Bank (2015)*.

As Eq. (1) indicates, the four determinants were assumed to have a significant positive relationship with ART coverage. In other words, countries with a higher HIV prevalence, higher income, higher level of health expenditure and a larger number of nurses were assumed to have higher levels of ART coverage. This conjecture concerning the determinants of ART coverage could be expressed in four hypotheses, which are based on

previous studies (*Lieberman, 2007*; *Nattrass, 2008*; *Schwardmann, 2008*; *Peiffer & Boussalis, 2010*; *Man et al., 2012*):

H1: *Countries with higher levels of HIV prevalence have a wider coverage of ART treatment among people living with HIV.*

H2: *Countries with higher levels of real income per person have a wider coverage of ART treatment among people living with HIV.*

H3: *Countries with higher levels of real health expenditure per person have a wider coverage of ART treatment among people living with HIV.*

H4: *Countries with larger numbers of nurses and midwives per one thousand people have a wider coverage of ART treatment among people living with HIV.*

To be more specific, the first hypothesis assumes that Sub-Saharan countries with a greater number of HIV patients would have higher levels of ART coverage because these countries would be compelled to make greater efforts to contain the HIV epidemic. There is also a temporal ordering in this relationship: countries with higher levels of ART coverage would have more HIV patients because more HIV-infected persons would stay alive due to the ART treatment (*Zaidi et al., 2013*). The second hypothesis assumes that Sub-Saharan African countries with higher levels of GDP per capita would have higher levels of ART coverages. This is because a nation's GDP per capita can be considered as a comprehensive economic indicator to measure general well-being of the people; this includes the provisions and access to medical services. In other words, Sub-Saharan African countries with higher GDP per capita are likely to provide better quality of medical care. These countries are more likely to have better hospitals, clinics and medical schools, more highly trained medical professionals, better medical equipment, more comprehensive implementation of public health programs and a higher awareness of the importance of hygiene. All of these are likely to influence the implementation of ART coverage. The third hypothesis assumes that countries with higher health expenditures would have higher levels of ART coverage because these countries could spend more money on providing a better medical infrastructure. The fourth hypothesis assumes that countries with greater numbers of nurses and midwives would have higher levels of ART coverage because these countries would have lager pools of human resources to roll-out ART treatment in clinics.

The data is analysed using both descriptive statistics and inferential statistics. The purpose of descriptive statistics analysis (i.e., the scatter plot analysis and the matrix scatter plot analysis) in this study was to visually examine the relationship between ART coverage and the four proposed determinants. A methodological advantage of the scatter plot analysis is that it allows visualising the relationship between the variables. The matrix scatter plot analysis allows capturing the structure of associations among the variables.

As to the inferential statistics, which included the correlation analysis and the multiple regression analysis, its main objective was to examine whether there were statistically significant associations between ART coverage and its four proposed determinants. A methodological advantage of the Pearson correlation analysis is that it can help to

determine whether ART coverage and its determinants had a statistically significant association in the context of bivariate estimations. In contrast, the subsequent multiple regression analysis was used to examine whether there were statistically significant associations between the variables in the context of multivariate estimations.

Two statistical packages, namely, the IBM SPSS Statistics Version 22 (IBM Corp., United States) and the EViews 8 for Windows (IHS Global Inc., United States), were used for the empirical analysis. To be more specific, the SPSS software aided with the descriptive statistics and inferential statistics analyses, namely, the matrix scatter plot analysis, the Pearson correlation analysis and the multiple regression analysis. The EViews software facilitated the scatter plot analysis.

## DESCRIPTIVE STATISTICAL ANALYSIS

First of all, the descriptive statistical analysis of the data, which involved the scatter plot method and the matrix scatter plot method, was applied. The scatter plot method helped to visualise relationships between ART coverage and the four hypothesised determinants (i.e., HIV prevalence, income level, medical expenditure and number of nurses). The matrix scatter plot analysis helped to capture general characteristics and relationships among ART coverage and the four determinants.

Figure 1 offers a visual representation of the relationship between antiretroviral therapy coverage (*ART*) and HIV prevalence (*HIV*). The *x*-axis is the percentage of people living with HIV in the total population between the ages 15 and 49. The *y*-axis is the percentage of people living with HIV who have access to antiretroviral therapy. The figures visually suggest the presence of a strong positive relationship between *ART* and *HIV*. This means that countries with a higher HIV prevalence tended to have a wider ART coverage. For example, in Botswana (*BWA*) the HIV prevalence was relatively high (22%) and its ART coverage was also relatively high (70%). A strong positive relationship among the variables also means that the Sub-Saharan African countries with a lower HIV prevalence tended to have a lower ART coverage. For instance, in Madagascar (*MDG*) the HIV prevalence was relatively low at 0.4% and the ART coverage was also relatively low at 1% . Figure 1 also indicates that there were several outliers. For example, Lesotho (*LSO*) had a high HIV prevalence (23%) but a relatively low ART coverage (28%). In contrast, the Gambia (*GAM*) had a relatively low HIV prevalence (4%) but a high ART coverage (56%).

Figure 2 offers a graphical representation of the relationship between antiretroviral therapy coverage (*ART*) and per capita income (*GDP*). The *x*-axis is the real per capita income denominated in thousand international dollars (I\$) and the *y*-axis is ART coverage. A visual inspection of the figure suggests the presence of a moderately strong positive relationship between *ART* and *GDP*, which may suggest that the Sub-Saharan African countries with lower income levels tended to have a lower ART coverage. For example, in the Central African Republic (*CAF*) the per capita income was low (I\$584), as was the ART coverage (14%). At the same time the Sub-Sharan countries with higher income levels tended to have a higher ART coverage. Thus, in Namibia (*NAM*) the per capita income was relatively high (I\$ 9,275) and the ART coverage was also high (52%). As can be seen

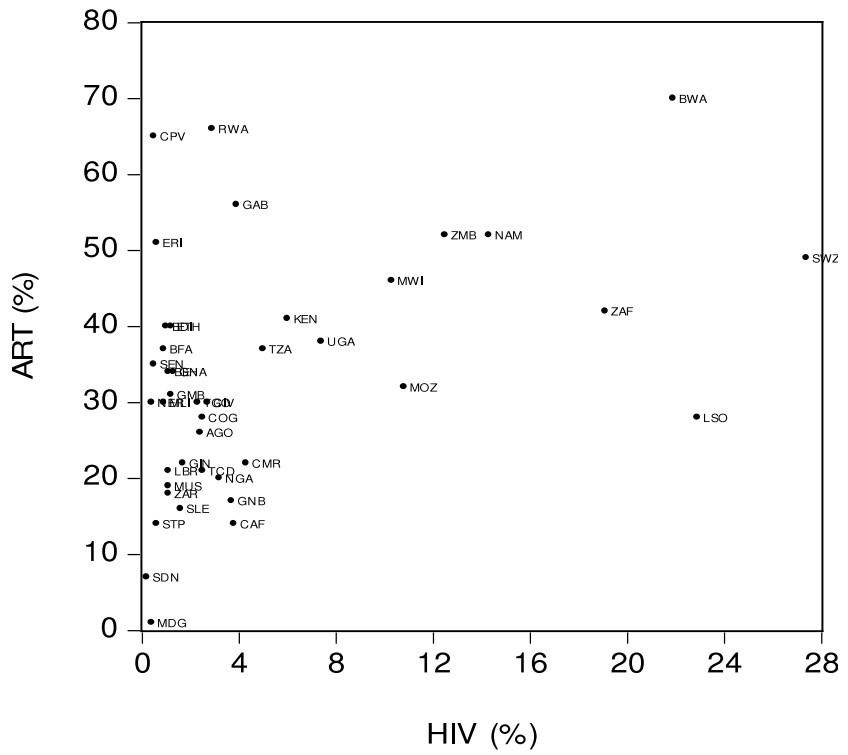

**Figure 1 Antiretroviral therapy coverage (ART) and HIV prevalence (HIV).** Notes: *ART* is the percentage of people living with HIV who have access to antiretroviral therapy. *HIV* is the percentage of people living with HIV in the total population aged 15–49. Source: *World Bank (2015)*.

in Fig. 2, there were more than a few outliers. For example, Angola (*AGO*) was one of the countries where income level was higher (I\$ 7,485) but the ART coverage was lower (26%). In contrast, in Rwanda (*RWA*) the per capita income was relatively low (I\$ 1,426) while the ART coverage was relatively high (66%).

Figure 3 visually displays the relationship between antiretroviral therapy coverage (*ART*) and per capita health expenditure (*HED*). The *x*-axis is the real per capita health expenditure denominated in thousand international dollars (I\$) and the *y*-axis is ART coverage. The figure suggests the presence of a moderately strong positive relationship between *ART* and *HED*. Thus, the countries with higher levels of health expenditure tended to have a higher ART coverage, and vice versa. For instance, in Swaziland (*SWZ*), a high per capita health expenditure (I\$563) was matched by a wide ART coverage (49%). By contrast, Liberia (*LDR*) had a low per capita income (I\$ 81) and its level of ART coverage was also low (21%). As this was the case with the previously reported findings, there were several outlier countries in the *ART–HED* relationship. Some countries had low levels of health expenditure but a relatively high ART coverage. For example, in Malawi (*MWI*), the per capita health expenditure was low (I\$ 90) while the level of ART coverage was high (46%). In contrast, Mauritius' (*MUS*) relatively high per capita health expenditure (I\$ 863) was not matched by a level of ART coverage (19%).

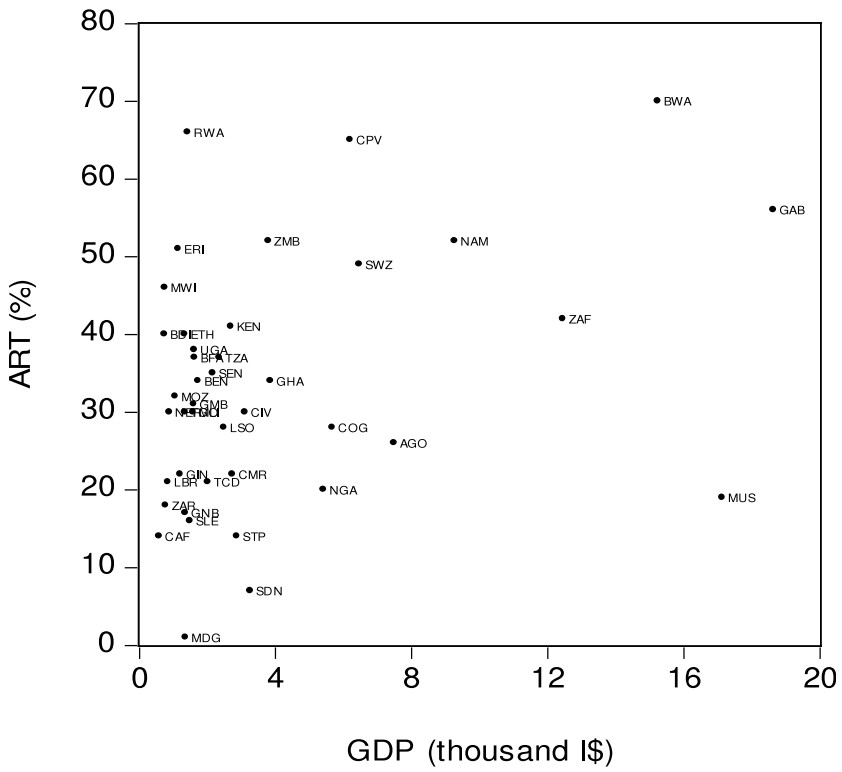

**Figure 2 Antiretroviral therapy coverage (ART) and per capita income (GDP).** *ART* is the percentage of people living with HIV who have access to antiretroviral therapy. *GDP* is income level or real GDP per person based on purchasing power parity (PPP) calculation. Source: *World Bank (2015)*.

Figure 4 depicts the relationship between antiretroviral therapy coverage (*ART*) and number of nurses and health care professionals (*NUR*). The *x*-axis is the number of nurses and midwives per one thousand people while the *y*-axis is ART coverage. A visual inspection of the figure suggests the presence of a weak positive relationship between *ART* and *NUR*. This means that countries with greater numbers of nurses and midwives tended to have wider ART coverages. For example, in South Africa (*ZAF*), the number of nurses and midwives per thousand people was relatively high (5.11 persons), as was the ART coverage (42%). Sierra Leone (*SLE*) was among the countries with lower numbers of nurses and midwives per thousand persons (0.16 persons) and lower ART coverage (16%).

Finally, the matrix scatter plot analysis helped to capture a general structure of the associations between the coverage and its four proposed determinants. Matrices showing all possible combinations among the five variables—*ART*, *HIV*, *GDP*, *HED* and *NUR*—are visualised in Fig. 5. An interesting insight gained from a visual inspection of the figure was the presence of very strong positive relationships between the economic variable (*GDP*) and the two public health variables (*HED* and *NUR*). This indicates that Sub-Saharan countries with higher national incomes also had higher health expenditures and higher numbers of nurses and midwives. Furthermore, a moderate positive relationship was found to exist between *HIV* and *GDP*. In other words, wealthier Sub-Saharan African countries tended to have more people living with HIV. It should be noted, however, that

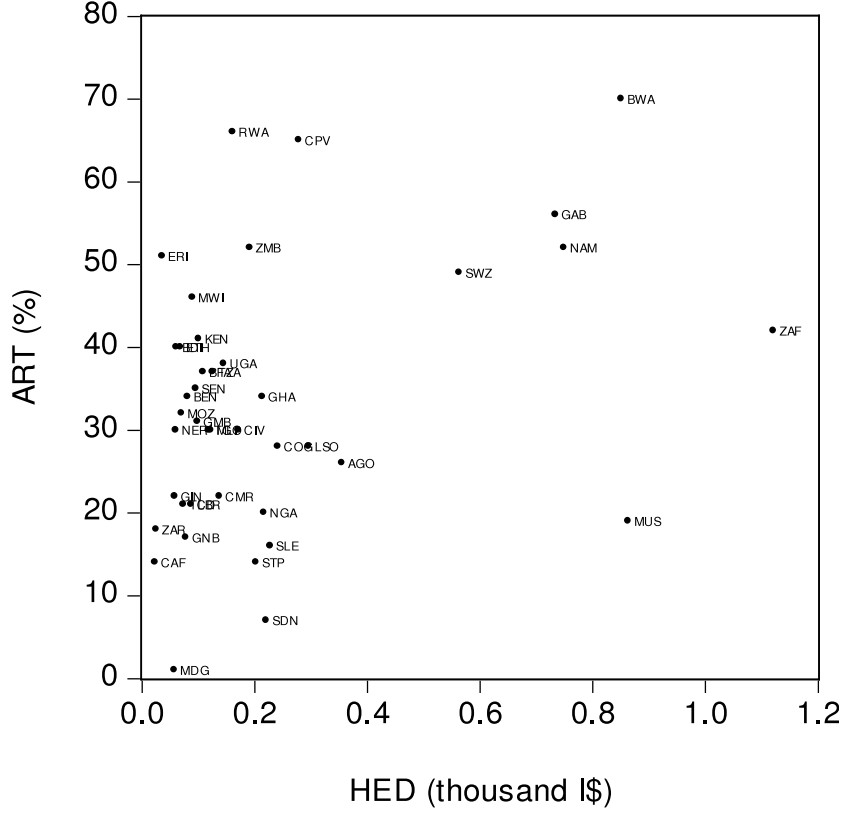

**Figure 3** **Antiretroviral therapy coverage (ART) and health expenditure (HED).** *ART* is the percentage of people living with HIV who have access to antiretroviral therapy. *HED* is real health expenditure per person based on PPP calculation. Source: *World Bank (2015)*.

there were more than a few outlier countries in this positive *HIV–GDP* relationship. Another finding was a strong positive association between *HIV* and *HED*. This implies that countries with higher HIV prevalence tended to have higher medical expenditures per person. An important finding was that, among the four proposed determinants, a strong association was found to exist between *ART* and *HIV*. However, this association was weaker than the association between *HIV* and *HED*.

## INFERENTIAL STATISTICAL ANALYSIS

This section reports findings from the inferential statistical analyses, namely, the correlation analysis and the multiple regression analysis. These tests examined whether there were statistically significant relationships between ART coverage in the selected 41 Sub-Saharan African countries and its four proposed determinants. The correlation analysis is based on bivariate estimations while the multiple regression analysis is based on multivariate estimations.

As seen in Table 1 which shows the findings of correlation analysis, there were nine statistically significant relationships between ART coverage and its determinants. First of all, ART coverage was found to have strong, positive and statistically significant associations with three of the four proposed determinants, namely, HIV prevalence,

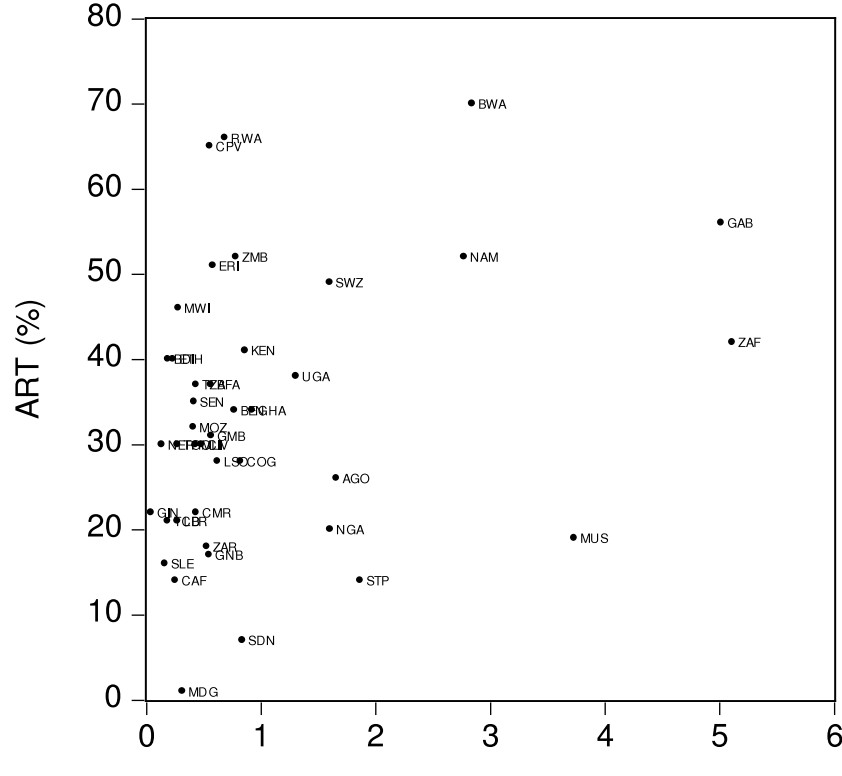

**Figure 4** **Antiretroviral therapy coverage (ART) and number of nurses (NUR).** *ART* is the percentage of people living with HIV who have access to antiretroviral therapy. *NUR* is number of nurses and midwives per one thousand people. Source: *World Bank (2015)*.

level of per capita income and level of per capita health care expenditure. The strongest association was found between ART coverage and HIV prevalence ($0.7 > r \geq 0.4$) and the correlation coefficient was statistically significant at 1% ($p < 0.01$). Further, ART coverage had moderately strong associations with income level and health expenditure level ($0.4 > r \geq 0.2$) and the correlation coefficients were statistically significant at 5% ($p < 0.05$). These results indicate that ART coverage in Sub-Saharan Africa was jointly determined by three elements, namely, HIV prevalence, level of national income per capita and level of health expenditure per capita.

Secondly, the HIV prevalence had statistically significant associations with three other determinants, namely, income level, health care expenditure and number of nurses and midwives. Among the three determinants, the strongest association was between the HIV prevalence and health care expenditure ($0.7 > r \geq 0.4$) and the correlation coefficient was statistically significant at 1% ($p < 0.01$). This means that the Sub-Saharan African countries with higher health expenditure tended to have a higher HIV prevalence. Also it was found that the HIV prevalence had a moderately strong association with income level ($0.4 > r \geq 0.2$); the correlation coefficient was statistically significant at 5% ($p < 0.05$). This means that countries with higher national incomes tended to have a higher HIV

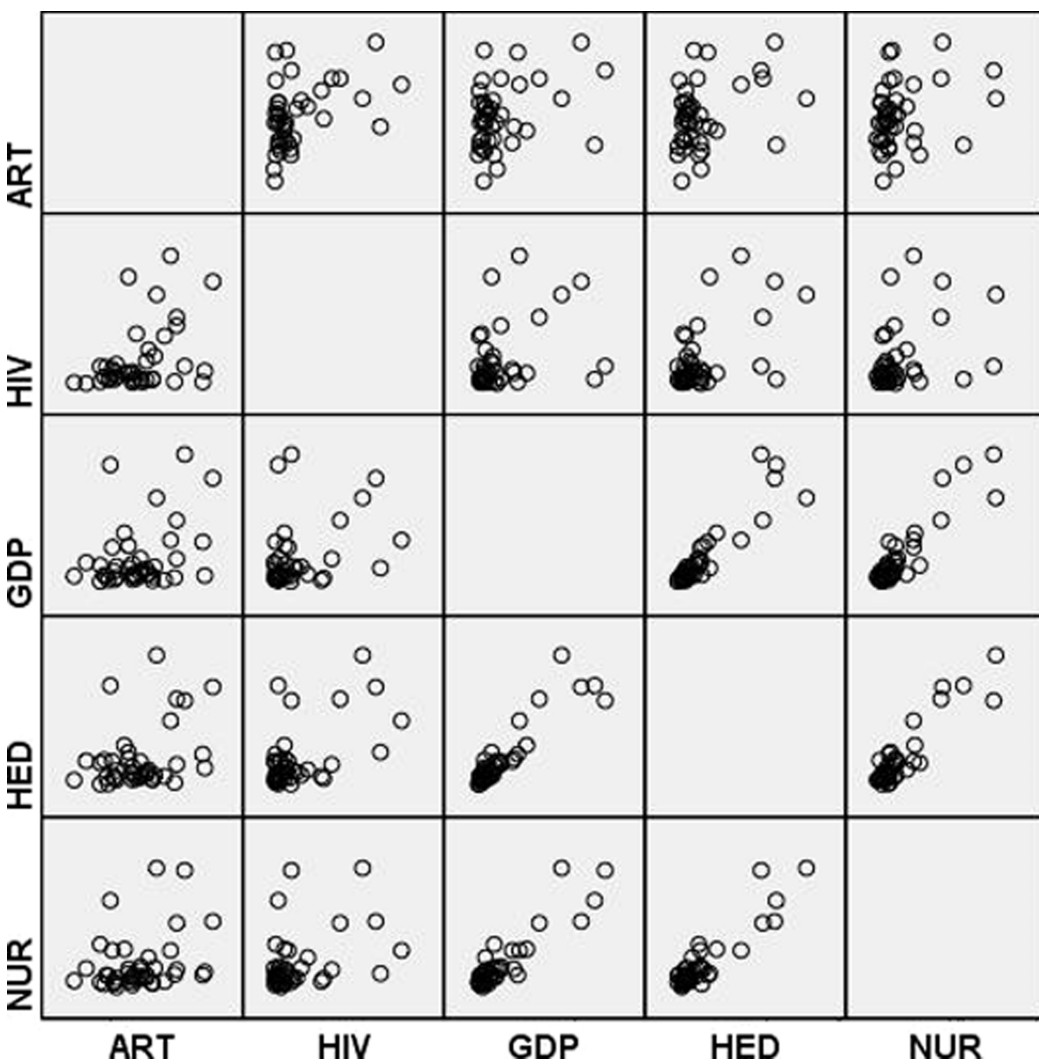

**Figure 5 Matrix scatterplot analysis.**

**Table 1 Correlation analysis.**

|  | ART | HIV | GDP | HED | NUR |
|---|---|---|---|---|---|
| ART | 1.000 |  |  |  |  |
| HIV | 0.417[**] | 1.000 |  |  |  |
| GDP | 0.353[*] | 0.354[*] | 1.000 |  |  |
| HED | 0.360[*] | 0.563[**] | 0.913[**] | 1.000 |  |
| NUR | 0.296 | 0.385[*] | 0.911[**] | 0.915[**] | 1.000 |

**Notes.**
[**] Denotes significance at the 0.01 level.
[*] Denotes significance at the 0.05 level.

prevalence. However, the positive *HIV–GDP* association was not strong and there were more than a few outliers in this relationship. In a similar way, the findings revealed a moderate association between the HIV prevalence and the number of nurses and midwives $(0.4 > r \geq 0.2)$; the correlation coefficient was statistically significant at 5% $(p < 0.05)$. It should be noted that while the number of nurses and midwives had a statistically significant association with the HIV prevalence, this was not the case concerning the ART coverage. This finding suggests that an increase in the HIV prevalence was followed by an increase in the number of medical professionals including nurses. However, the increased number of nurses was not accompanied by an increase in ART coverage.

Thirdly, the level of national income had statistically significant associations with two other determinants, namely, health care expenditure and number of nurses and midwives $(r \geq 0.7)$; the correlation coefficients were statistically significant at 1% $(p < 0.01)$. This finding indicates that relatively wealthy Sub-Saharan African countries had higher expenditures on health care and they had higher numbers of employed nurses and midwives. At the same time, the association between income level and ART coverage was much weaker than the associations between the income level and the two public health related variables (i.e., health care expenditure per capita and number of nurses and midwives). This means that while wealthier Sub-Saharan African countries were spending larger amounts of money on health care as a whole, these expenses were not matched by the expenditure on ART coverage. Furthermore, while the relatively wealthy African countries tended to employ more medical professionals, including nurses and midwives, this did not entail the higher levels of ART coverage. This means that medical professionals in these countries were likely to be involved in general health care provision rather than in ART programs.

Fourthly, the findings revealed that a statistically significant and very strong association existed between the level of health expenditure and the number of nurses and midwives $(r \geq 0.7)$; the correlation coefficient was significant at 1% $(p < 0.01)$. This means that higher health expenditures tended to have a statistically significant association with higher numbers of nurses and midwives. This fact may suggest that relatively wealthy Sub-Saharan African countries could afford to employ more health care professionals, such as nurses and midwives.

In the next stage, the multiple regression analysis examined whether there were meaningful and statistically significant relationships between ART coverage and its four determinants. For this purpose, four slope coefficients in the multiple regression analysis were used to test the four hypotheses put forward earlier in this study regarding the relationship between ART coverage and its four determinants. Table 2 shows the findings from multiple regression analysis. As can be seen in the table, the *R*-squared was equal to 0.243. This means that 24.3% of variance in the ART coverage could be explained by the four determinants.

Furthermore, the slope coefficient for HIV prevalence was 1.001 and it was significant at the 5% level. The significance of this slope coefficient implies that there was a statistically significant and positive relationship between *ART* and *HIV*. In other words, the level of HIV prevalence could be considered as a determinant of the level of ART coverage.

**Table 2  Multiple regression analysis.**

|  | Unstandardized coefficient *B* | Standard error | Standardised coefficient *Beta* | *t*-statistics |
|---|---|---|---|---|
| Constant | 26.040 | 3.210 |  | 8.112[**] |
| *HIV* | 1.001 | 0.473 | 0.434 | 2.119[*] |
| *GDP* | 2.225 | 1.508 | 0.635 | 1.476 |
| *HED* | −19.607 | 32.180 | −0.326 | −0.609 |
| *NUR* | −1.965 | 5.395 | −0.151 | −0.364 |
|  | *R*-square 0.243 |  | Adjusted *R*-square 0.159 |  |

**Notes.**
[**] Denotes significance at the 0.01 level.
[*] Denotes significance at the 0.05 level.

The slope coefficient for another variable, *GDP*, was 2.225 and it was not statistically significant. This means that there was a positive but not statistically significant relationship between *ART* and *GDP*. Next, the slope coefficient for *HED* was estimated at −19.607, and it was not significant. Similarly, the slope coefficient for *NUR* was −1.965, and it was not significant. These findings imply that there were negative and not statistically significant relationships between the ART coverage and the two public health-related variables, namely, level of health expenditure and number of nurses and midwives per thousand people.

Importantly, statistically significant slope coefficient for HIV prevalence (*HIV*) substantiated Hypothesis One of this study. It proposed that higher levels of HIV prevalence would result in a wider coverage of ART treatment among people living with HIV. On the other hand, statistically non-significant slope coefficient for the per capita income (*GDP*) refuted Hypothesis Two, which postulated that countries with higher levels of real income per person would have a wider ART coverage.

Similarly, statistically non-significant slope coefficients for the two public health-related variables—*HED* and *NUR*—refuted Hypothesis Three and Hypothesis Four. The former proposed that countries with higher levels of real health expenditure per person would have a wider ART coverage. The latter assumed that countries with larger numbers of nurses and midwives per one thousand people would have a wider ART coverage.

Thus, among the four hypotheses put forward in this study, the findings from the multiple regression analysis provided empirical evidence only in support of Hypothesis One. In short, the empirical findings indicated that among the four proposed determinants only the level of HIV prevalence had a statistically significant impact on ART coverage in Sub-Saharan African countries.

## DISCUSSION

By and large, the findings of the current study agree with results reported in earlier investigations. This means that some consistent conclusions have been reached concerning determinants of ART coverage. To be more specific, the present study found that there was a significant positive relationship between HIV prevalence and ART coverage in the 41 Sub-Saharan African countries. Among the earlier empirical investigations, five

**Table 3  Multiple regression analysis (without Rwanda).**

|  | Unstandardized coefficient B | Standard error | Standardized coefficient Beta | t-statistics |
|---|---|---|---|---|
| Constant | 24.479 | 2.949 |  | 8.300[**] |
| HIV | 1.120 | 0.429 | 0.514 | 2.610[*] |
| GDP | 2.850 | 1.380 | 0.859 | 2.066[*] |
| HED | −28.428 | 29.250 | −0.500 | −0.972 |
| NUR | −2.360 | 4.881 | −0.192 | −0.483 |
|  | R-square 0.324 |  | Adjusted R-square 0.247 |  |

**Notes.**

[**] Denotes significance at the 0.01 level.

[*] Denotes significance at the 0.05 level.

studies (*Lieberman, 2007*; *Nattrass, 2008*; *Schwardmann, 2008*; *Peiffer & Boussalis, 2010*; *Man et al., 2012*) have focused on the relationship between HIV prevalence and ART coverage. One of them concluded that there existed a significant relationship between the two variables (*Nattrass, 2008*). Two studies (*Schwardmann, 2008*; *Peiffer & Boussalis, 2010*) detected the presence of a significant relationship between HIV and ART but failed to find an overall significant relationship between these variables. It should be noted that the three studies (*Nattrass, 2008*; *Schwardmann, 2008*; *Peiffer & Boussalis, 2010*) employed different estimation models to analyse ART determinants. As a result, the study by *Nattrass (2008)* found that HIV prevalence was consistently significant in all three models, while the studies by *Schwardmann (2008)* and *Peiffer & Boussalis (2010)* discovered that HIV prevalence was significant only in the estimation models with a fewer number of determinants. The remaining two empirical investigations (*Lieberman, 2007*; *Man et al., 2012*) concluded that the relationship between HIV prevalence and ART coverage was positive but non-significant. The findings of the current and previous studies point to some similarities in the relationship between *ART* and *HIV* in the 41 Sub-Saharan African countries and other developing countries.

Secondly, the current study has detected a positive but non-significant relationship between ART coverage and GDP per capita in Sub-Saharan African countries. All of the previous studies that examined the *ART–GDP* relationship reported a positive significant association between these variables. The discrepancies in the findings could be due to the presence of outlier countries in this study. For example, Rwanda is a relatively poor country with per capita income of I\$ 1,426. However, its ART coverage at 66% is among the highest in the region. To confirm this proposition, Table 3 shows results of the multiple regression analysis that excluded Rwanda. As can be seen in the table, the *R*-squared increased from 0.243 in the original estimation model (presented in Table 2) to 0.324 in the new estimation model. More importantly, other determinants of ART coverage, namely *HIV* and *GDP*, were found to be statistically significant. This means that when an outlier country was excluded from the analysis, the findings concerning existence of a statistically significant relationship between ART coverage and national income were in line with findings reported in the earlier investigations.

Thirdly, the findings revealed that there was a positive but non-significant relationship between ART coverage and health care expenditure per capita in the 41 Sub-Saharan African countries. Only two of the available studies (*Lieberman, 2007*; *Schwardmann, 2008*) have focused on the *ART–HED* relationship and the findings are contradictory. For example, *Lieberman (2007)* found a positive but non-significant relationship between *ART* and *HED*, while *Schwardmann (2008)* detected a positive and significant relationship. The findings of the current study are in line with those reported by *Lieberman (2007)*. The discrepancies in results could be attributed to the differences in measuring health care expenditure. The current study used a direct method to measure health care expenditure and it relied on a monetary value denominated in International dollar. By contrast, *Schwardmann (2008)* employed an indirect method and relied on the share of GDP spent on health care.

Finally, the current study detected a positive but non-significant relationship between ART coverage and number of nurses and midwives per one thousand persons. Similar results were achieved by two previous investigations (*Nattrass, 2008*; *Schwardmann, 2008*) on the relationship between ART coverage and number of health care professionals. Both of the studies reported a positive but non-significant relationship between the variables.

## CONCLUSION

There is a lack of studies on determinants of ART coverage in developing countries. Obviously, a wider and deeper research on this topic is much needed because, despite considerable global efforts to achieve universal ART coverage, only one-third of the people living with HIV have access to the treatment. This preliminary study investigated determinants of ART coverage in Sub-Saharan Africa. It discovered that among the four proposed determinants only HIV prevalence had a statistically significant impact on ART coverage. A novelty in this paper is that it is the first empirical investigation that focuses on Sub-Saharan African countries and systematically examines determinants of ART coverage in this geographical area. The majority of earlier empirical analyses of the relationship between HIV and ART (*Lieberman, 2007*; *Nattrass, 2008*; *Schwardmann, 2008*; *Peiffer & Boussalis, 2010*; *Man et al., 2012*) failed to detect a significant relationship between the two variables, which appears counter-intuitive. The present study was able to conclude that the level of HIV prevalence had been a significant factor to affect the provision of ART coverage in Sub-Saharan Africa.

There are several shortcomings in this study. Firstly, the data were at national levels and thus they did not reflect differences among regions within the countries. To overcome this limitation, future investigations may want to employ more detailed data compiled by national statistics departments. Secondly, only four determinants of ART were considered in this study and the sample size was forty-one. A greater number of ART determinants and wider geographical areas could be investigated in future. Such studies will allow a better understanding of issues related to health care provision to the people living with HIV, which could lead to a higher equality of access to health care services and help to achieve the ultimate goal of health for all.

### Funding

The authors received no funding for this work.

### Competing Interests

The authors declare there are no competing interests.

### Author Contributions

- Fumitaka Furuoka analyzed the data, wrote the paper, prepared figures and/or tables.
- Mohammad Zahirul Hoque reviewed drafts of the paper.

### Data Availability

Fumitaka Furuoka's Webpage

https://sites.google.com/site/fumitakafuruokaswebpage/data-and-oxgauss-codes/paper-11.

### Supplemental Information

Supplemental information for this article can be found online at http://dx.doi.org/10.7717/peerj.1496#supplemental-information.

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
