# Peer review of "Determinants of antiretroviral therapy coverage in Sub-Saharan Africa"

_PeerJ, doi:10.7717/peerj.1496_

## Round 0.1 · original submission · Minor Revisions

Please address all of the comments made by the reviewers, then it should be acceptable for publication.

Reviewer 1 ·

Basic reporting

The paper is very comprehensive and the details of the methods are explained clearly and thoroughly. This clarity makes the paper accessible and pleasing to read. However, sometimes the authors slip into overly explaining minute details of their data – the simple scatter plot of Figure 1 is explained over three paragraphs, describing individually 8 points from the 41-point graph. The authors would benefit from some more brevity in the overly loquacious descriptions in the paper.

The authors would benefit from having the manuscript proofed by a native English speaker. Overall the grammar is good, but there is some awkwardness that leads to more difficult reading comprehension. One main example is overuse of 'the' - 'the ARV treatment' is almost always just 'ARV treatment.'

Experimental design

No comment.

Validity of the findings

I have no reservations about the paper being fit to publish in a journal like PeerJ -- the analysis may be of interest and is scientifically and methodologically sound overall. However, one of my major concerns in the paper is that the authors tend to overstep in speculation what is actually supported by the analysis done and the results obtained:

1. In the last paragraph on page 5, the authors should be cautious about stating things like ‘Figure 1 shows there was a strong positive relationship between ART and HIV.’ At this point no statistical analysis has been done, so stating that a relationship is present is a bit premature. It would be safer to say that the figure visually suggests the presence of a relationship, which was confirmed in the statistical analysis below. The same applies to figure 2. Also to figure 5, the matrix scatter plot analysis.

2. In the second paragraph on page 9, the authors state, “In view that the HIV prevalence had a statistically meaningful positive association with the health expenditure, the Sub-Saharan African countries with larger populations living with HIV may want to increase their expenditures for health care.” I think the authors should remove this statement, as it’s quite misleading. No analysis has been done to look into causation, but it seems most logical to assume that the high HIV prevalence is what is driving the association with high health care expenditure – so it doesn’t make much sense to ask countries with high HIV prevalence to raise their health care expenditure further.

3. Also in the second paragraph on page 9, the authors write, “This finding indicates that an increase in the HIV prevalence was followed by an increase in the numbers of medical professionals including nurses.” The authors should be careful here – there is no causation analysis, so they should perhaps substitute ‘This finding suggests’ rather than ‘indicates,’ as no other evidence is given to support the idea that the increase in HIV prevalence lead to an increase in nurses.

4. Similarly, in the second paragraph on page 10, the authors write about the existence of a ‘virtuous circle,’ where spending on healthcare leads to more nurses and midwives, which leads to better healthcare, which leads to more spending on healthcare. I think from the data presented, there’s little evidence of this more complex circle of causation – all the correlation shows is that more healthcare spending is associated with more nurses and midwives. I think it is too far of a causal leap to assume there is anything ‘circular’ about this, as the relationship could easily be quite linear, and either spending more on healthcare could lead to more nurses, or the decision to employ more nurses could lead to more spending on healthcare.

Additional comments

I have some general comments below:

5. Authors give good reasons for sticking to a lower number of more easily definable and measurable elements to include in the model. However, they do not explain why they leave out easily-definable elements that were found to be important in 2 or more of the aforementioned studies, ex: foreign aid levels and number of HIV patients in urban areas. Perhaps the data is simply not available?

6. Please check citation for reference 16 – I cannot locate it from the citation given.

7. The ‘paper organization’ paragraph included as the last paragraph of the introduction is unnecessary.

8. The explanation of the equation for ART coverage and the conversion rates for the GDP and HED are clear and concise.

9. Figure axes need to have the units labelled. In figure 1 – both HIV prevalence and ART coverage should be shown as being percent (%). In Figure 2, the units are much less obvious – is GDP in thousands of dollars? Hundreds of dollars? Just dollars? The authors need to be much more clear with the units in the figures.

10. Also, though the description of the figures in the text body is good, this description should be also present in the figure legends, so that when viewing the figures, a full description of the axes, units, etc, are available very nearby.

11. It seems figures 6-8 should be tables rather than figures.

12. In the first paragraph of the discussion, the authors compare their findings to those of previous papers. They state that references 13, 14, and 15 also found a significant positive relationship between ART and HIV prevalence. However, when they outline the findings of these papers at the beginning of their introduction, they only state that reference 14 found HIV prevalence significantly influenced ART coverage. This easily leads to confusion as the two paragraphs seem to contradict each other. I suggest the authors include everything the previous papers found significant in their introduction to avoid this.

Reviewer 2 ·

Basic reporting

The study deals with an important topic and is relatively
well written and clearly explained.

1) The authors give references [7-11] for the
effectiveness of ART on survival rate of
HIV-infected persons. The Cohen 2011 reference is
important, but more recent and relevant references
can be given. Please cite:

Tanser, Frank, et al. "High coverage of ART
associated with decline in risk of HIV acquisition
in rural KwaZulu-Natal, South Africa." Science
339.6122 (2013): 966-971.

Vandormael, Alain, et al. "Use of antiretroviral
therapy in households and risk of HIV acquisition
in rural KwaZulu-Natal, South Africa, 2004–12: a
prospective cohort study." The Lancet Global
Health 2.4 (2014): e209-e215.

Both these studies specifically explore the role
of ART coverage and HIV prevalence on HIV
acquisition risk in the sub-Saharan context.

2) The study only explores four determinants,
which is a small number. There is agreement that
other determinants may not be easily measured, but
they are still important and this should be
acknowledged, rather than so quickly dismissed as
the authors seem to do. For example, the cases of
South Africa under T. Mbeki and Uganda under Y.
Museveni show how these two differing political
responses had a profound impact on national policy
and the roll-out of ART in their respective
countries. The study is someone limited, not
strengthened, by not being able to include these
factors in a statistical analysis.


3) It should be noted that the data used, World
Development Indicators, contains measures that are
themselves estimates and approximations. The data
is at a national level, and will mask
heterogeneity within countries that relate to
differences in ART coverage and health-care
expenditure by province or district etc. The
potential limitations of the dataset should also
be acknowledged.


4) The four hypotheses: The authors need to be far
more explicit in outlining the causal mechanisms
and pathways between the variables in each of the
hypothesis.

H1: Temporal ordering of this relationship needs
to be acknowledged. The reverse of the hypothesis
is equally likely. Countries that have a higher
ART coverage will see a higher HIV prevalence,
since more people with HIV will live longer if
they have access to ART. See:

Zaidi, Jaffer, et al. "Dramatic increases in HIV
prevalence after scale-up of antiretroviral
treatment: a longitudinal population-based HIV
surveillance study in rural kwazulu-natal." AIDS
(London, England) 27.14 (2013): 2301.

H2-4) Please explain why there should be
associations between the variables in these
hypotheses. Why should higher levels of income per
person result in higher ART coverage. In some
countries, ART is provided free of charge. I am
struggling to see the specific link here. What is
the nurses/midwives measure trying to capture? The
ability of a government to roll-out treatment
through its health-clinics? Please explain.


5) Is the HED a measure of expenditure on HIV
treatment and care specifically or just health
generally? How do we know what proportion of this
amount is going to HIV-infected individuals?

6) Authors should make an attempt to format their
results into proper tables, rather than just
submitting raw SPSS output.

Experimental design

1) The analysis shows the determinants are
correlated. Importantly, there is a positive
correlation between GDP and HIV prevalence. So
'wealthier' countries are likely to have a higher
prevalence, according to these results. To what extent
does this relationship mediate the results of the other
associations. For example, on page 6 the authors
write 'Countries with lower incomes have lower ART
coverage'. But this may simply be because they
have lower prevalence, thus decreasing the
necessity for ART spending.

2) The authors find that high ART coverage is
associated with a high HIV prevalence. This
result seems to be intuitive, since we would
expect countries with higher HIV prevalence to
react by providing more ART at public health-care
facilities. It is unclear what new information
this result really adds to what we already know as
of 2015.

3) There is a lot of multi-collinearity between
the variables of the study. How appropriate is it
therefore to include all of these determinants
together in a multi-regression model? Can the
authors regress HIV prevalence on each of the
determinants separately and show the results. It
could be that significant effects between each of
the determinants and the outcome are being washed
away when included together in one model.

4) Overall, I am not convinced that the
relationships between the determinants and each
other are acyclic, or between the determinants and
the outcome are acyclic. As the authors note in
the discussion, there is a ‘virtuous circle' cycle
going on here, and I don't believe the paper has
yet to convince me otherwise.

Validity of the findings

Please see above comments

·

Basic reporting

No comments

Experimental design

The authors described the study designs in details, however, the study sample size is relative small (N=41 countries) and only 4 factors were assessed in the study. The authors assumed that "only four elements have a decisive impact on the ART coverage in Sub-Saharan Africa", but they did not provide any evidence to support their assumption. An imporant factor, foreign aid, which is relevant to the study countries and the outcome and can also be assessed objectively, was not include in the study. In addition, the authors did not provide the reason why the number of nurses and midwives/1000 people, not the number of total medical professionals/1000 people, was used in the study.

Validity of the findings

The statistical analyses are appropriate, but again, given the small sample size, although the coefficients for GDP and HED are strong, they did not reach the significance level. Additionally, the r-square seems small (0.243), suggesting that other factors contributing to the ART coverage are not included in the study. The authors should correct the slope coefficient for the HIV prevalence (1.001, not 2.225) in the results.

Additional comments

The authors should add a paragraph of study limitations in the discussion. The length of the article should also be reduced as the detailed interpretation of the results is not necessary.

---

## Round 0.2 · accepted · Accept

Thank you for taking into account the input from the reviewers. The paper is now acceptable for publication.